# An Inducible ESCRT-III Inhibition Tool to Control HIV-1 Budding

**DOI:** 10.3390/v15122289

**Published:** 2023-11-22

**Authors:** Haiyan Wang, Benoit Gallet, Christine Moriscot, Mylène Pezet, Christine Chatellard, Jean-Philippe Kleman, Heinrich Göttlinger, Winfried Weissenhorn, Cécile Boscheron

**Affiliations:** 1University Grenoble Alpes, CEA, CNRS, Institut de Biologie Structurale (IBS), 38000 Grenoble, France; haiyan.wang@ibs.fr (H.W.); benoit.gallet@ibs.fr (B.G.); christine.chatellard@univ-grenoble-alpes.fr (C.C.); jean-philippe.kleman@ibs.fr (J.-P.K.); 2University Grenoble Alpes, CEA, CNRS, ISBG, 38000 Grenoble, France; christine.moriscot@ibs.fr; 3University Grenoble Alpes, INSERM, IAB, 38000 Grenoble, France; mylene.pezet@inserm.fr; 4Department of Molecular, Cell and Cancer Biology, University of Massachusetts Chan Medical School, Worcester, MA 01605, USA; heinrich.gottlinger@umassmed.edu

**Keywords:** HIV-1, budding, ESCRT-III, CHMP2A, CHMP3, CHMP4B, membrane remodeling, membrane fission, VLP release modulation

## Abstract

HIV-1 budding as well as many other cellular processes require the Endosomal Sorting Complex Required for Transport (ESCRT) machinery. Understanding the architecture of the native ESCRT-III complex at HIV-1 budding sites is limited due to spatial resolution and transient ESCRT-III recruitment. Here, we developed a drug-inducible transient HIV-1 budding inhibitory tool to enhance the ESCRT-III lifetime at budding sites. We generated autocleavable CHMP2A, CHMP3, and CHMP4B fusion proteins with the hepatitis C virus NS3 protease. We characterized the CHMP-NS3 fusion proteins in the absence and presence of protease inhibitor Glecaprevir with regard to expression, stability, localization, and HIV-1 Gag VLP budding. Immunoblotting experiments revealed rapid and stable accumulation of CHMP-NS3 fusion proteins. Notably, upon drug administration, CHMP2A-NS3 and CHMP4B-NS3 fusion proteins substantially decrease VLP release while CHMP3-NS3 exerted no effect but synergized with CHMP2A-NS3. Localization studies demonstrated the relocalization of CHMP-NS3 fusion proteins to the plasma membrane, endosomes, and Gag VLP budding sites. Through the combined use of transmission electron microscopy and video-microscopy, we unveiled drug-dependent accumulation of CHMP2A-NS3 and CHMP4B-NS3, causing a delay in HIV-1 Gag-VLP release. Our findings provide novel insight into the functional consequences of inhibiting ESCRT-III during HIV-1 budding and establish new tools to decipher the role of ESCRT-III at HIV-1 budding sites and other ESCRT-catalyzed cellular processes.

## 1. Introduction

HIV-1 assembly and budding take place at the plasma membrane [1,2] and require the interplay of viral structural proteins [3] and cellular factors to release new infectious virions [4,5,6,7,8]. Notably, budding is mostly driven by the polyprotein Gag, and its expression suffices to produce virus-like particles (VLPs) [9]. The p6 domain of Gag contains essential late domains [10,11,12,13] that have been shown to recruit the cellular endosomal complex required for transport (ESCRT) machinery via Tsg101, a subunit of ESCRT-I [14,15,16], and/or via the ESCRT-associated factor Alix [17,18,19,20].

The ESCRT machinery is composed of five complexes, ESCRT-0, -I, -II, -III, and VPS4 [21], which catalyze numerous membrane remodeling processes including topologically similar inside out budding implicating ESCRT-III and VPS4 in the final cut via membrane fission [22,23,24,25]. Mammalian cells express eight different ESCRT-III proteins, CHMP1 to 8, with CHMP1, CHMP2, and CHMP4 present in two and three isoforms, respectively [23]. ESCRT-III proteins shuttle between an inactive autoinhibited closed conformation [26,27] and an activated state [28,29,30] that polymerizes in the open ESCRT-III conformation [31,32,33,34], assembling into loose filaments or tight helical structures in vitro [27,35,36,37,38,39,40] and in vivo [41,42,43,44,45]. Most ESCRT-III proteins contain MIT-domain interacting motifs (MIMs) present within their C-termini recruiting VPS4 [46]. VPS4 was suggested to constantly remodel ESCRT-III in vivo [47], consistent with VPS4-catalyzed ESCRT-III filament remodeling in vitro [48,49]. Notably CHMP2A-CHMP3 filament remodeling into dome-like end-caps was suggested to constrict membrane necks from fixed diameters of 45 to 55 nm down to the point of fission [34,48,50].

Modified ESCRT-III (C-terminal deletions without MIM) or ESCRT-III fusion proteins as well as catalytic inactive VPS4A or B act in a dominant negative way and block HIV-1 budding upon over expression [17,18,19,28,51].

SiRNA depletion experiments suggested that HIV-1 budding requires, in principle, only one CHMP4 and one CHMP2 isoform to facilitate egress, whereas CHMP4 filament assembly provides a platform for downstream CHMP2 recruitment [52]. However, CHMP3 cooperates with CHMP2A to increase budding efficiency substantially, while CHMP2B acts independent of CHMP3 [53].

Live cell imaging of budding sites showed that Gag recruitment to the plasma membrane and VLP formation is, on average, completed within ~10 min [54]. Alix and Tsg101 (ESCRT-I) progressively accumulate with Gag [55], and once Gag recruitment is terminated, ESCRT-III proteins and VPS4 transiently appear at the budding site, followed by virus release [55,56]. Gag colocalizing ESCRT-III clusters showed closed, circular structures with an average size of 45 to 60 nm [57]. Recruitment is sequential, with CHMP4B arriving before CHMP2A followed by VPS4. This was suggested to constrict the neck, and in case scission does not occur within minutes after ESCRT-III remodeling and disassembly, ESCRT-III and VPS4 are recruited again to the same budding site [58]. The residence time of ESCRTs at the budding site is very short, a few minutes or less [59], and only 1 to 3% of total budding sites per cell exhibit ESCRT-III and Gag colocalization [57].

In order to facilitate imaging, which is challenged by the transient nature of ESCRT-III function, particularly in a context where cell physiology is not excessively compromised, we developed a drug-inducible transient inhibitory ESCRT-III system. To this end, CHMP2A, CHMP3, and CHMP4B (referred to collectively as CHMP) proteins were fused to the hepatitis C virus protease NS3 (NS3) via a short linker containing the NS3 cleavage site, which permits autocleavage (see Appendix A). Treatment of cells with the NS3 inhibitor Glecaprevir would induce the temporal accumulation of full-length CHMP-NS3-FP proteins. This accumulation potentially exerts a dominant-negative effect on ESCRT-III function, as previously reported when a heterologous protein is fused to CHMP proteins (Figure 1) [17,18]. We find that inducible CHMP3-NS3-FP had no effect on VLP release in contrast to CHMP3-YFP fusions, but increased the inhibitory effect in combination with CHMP2A-NS3-FP. Furthermore, we demonstrate that inducible CHMP4B-NS3-FP fusions and CHMP2A-NS3-FP fusion proteins can be fine-tuned to reduce Gag-VLP release and temporally increase the number of budding sites, revealing Gag-ESCRT colocalization.

## 2. Materials and Methods

### 2.1. DNA Constructs

Homo sapiens CHMP2A, CHMP3, and CHMP4B were fused in frame with an NS3 cleavage site (wild-type or mutated) and the NS3 Hepatitis C protease. They were also tagged with a Flag tag and with either mNeonGreen or mTurquoise (referred to as green and blue, respectively, or collectively called FP) and subsequently cloned into pCDNA3.1 (for protein sequences, please refer to Appendix A). The mNeonGreen and mTurquoise constructs were previously described [60,61]. Vps4B R253A, NS3, CHMP2A, CHMP3, and CHMP4B were synthesized (Life Technologies, Courtaboeuf, France). Other plasmids used in this study were previously reported, including pCG-GagRevInd-7ires-puro (referred to as Gag) [62,63], Gag-mCherry [1], pcDNA-Vphu (ARP-10076, NIH AIDS Reagent Program), GFP-Vps4A E228Q [64], and GFP-p40Phox [65].

### 2.2. Cells Culture, Transfection and Immunoblotting Assay

HeLa cells (ATCC, CCL-2), HEK-293T cells (ATCC, CRL-3216), and Hela Kyoto cells (Bst2+ or Bst2−) [66] were cultured in high-glucose DMEM (Life Technologies, Courtaboeuf, France) supplemented with 10% FBS (Life Technologies, Courtaboeuf, France) and L-glutamine (2 mM, Sigma Aldrich, St. Quentin Fallavier, France). They were maintained at 37 °C in a humidified incubator with 5% CO_2_. FreeStyle293F cells (Life Technologies, Courtaboeuf, France) grown in FreeStyle293 Expression Medium (Life Technologies, Courtaboeuf, France) were maintained at 37 ℃ with 8% CO_2_ on an orbital shaker.

For immunoblotting analysis, cells were seeded at 60% confluence into 100 mm dishes and transfected 24 h later using the Jetprime technique (Ozyme, Saint Cyr l’Ecole, France). The cells were cotransfected with 0.5 μg of Gag, the specified amount of CHMP-NS3-FP, and a total transfected DNA amount of 8 μg. Glecaprevir (CliniSciences, Nanterre, France) was used at a concentration of 25 μM.

Whole cells and VLPs protein extracts were prepared as previously described [64]. Western blots were conducted using the following antibodies: Anti-HIV-1 p24 (ARP-3537, NIH AIDS Reagent Program), Anti-HIV-1 Vpu (ARP-969, NIH AIDS Reagent Program), Anti-Bst2 (ARP-11721, NIH AIDS Reagent Program), anti-GFP (#A11122, Life Technologies, Courtaboeuf, France), and Anti-Flag (#F7625, Sigma Aldrich, St. Quentin Fallavier, France). Quantification was based on densitometry comparing Gag detection in the VLP fraction to total Gag within cells. To additionally correct for Gag intensities, densitometry values were normalized for Gag to 1 (by dividing the raw mean values by those measured in Gag).

For live cell imaging, cells were seeded in fibronectin-coated glass-bottomed μ-dishes (Biovalley, Nanterre, France). HeLa CCL2 cells beyond passage p15 and HeLa Kyoto BST2– cells were transfected with an excess of an untagged version of Gag to prevent the morphological defects associated with particles assembled solely from fluorescently tagged Gag [67]. Cells were cotransfected with 0.4 μg of pGag, 0.1 μg of pGag-mCherry, and either 0.5 μg of GFP-VPS4A_E228Q, 0.5 μg of GFP-VPS4B_R253A, or 0.5 μg of pCHMP-NS3-FP. Fluorescent imaging of live cells was conducted 24 h after transfection, following a 4 h treatment with Glecaprevir (25 µM, CliniSciences, Nanterre, France) or an equivalent volume of DMSO. An exception was made in the case of the HeLa CCL2 TIRF video-microscopy experiment, where Glecaprevir incubation was limited to 2 h.

For electron microscopy imaging, FreeStyle293F cells were seeded in a 6-well plate (Life Technologies, Courtaboeuf, France) at a density of 2 × 105 cells per well. After 24 h, the cultures were cotransfected with 2 µg of pCHMP-NS3-Flag and 2 µg of pGag using 4 µL of 293Fectin Reagent (Life Technologies, Courtaboeuf, France). Cells were pelleted and high-pressure frozen as described in [68] using an HPM100 system (Leica Microsystems, Vienna, Austria). After freezing, samples were cryo-substituted in an AFS2 machine (Leica Microsystems, Vienna, Austria), dehydrated, and embedded in anhydrous Araldite resin.

### 2.3. Image Acquisition and Analysis

Spinning disc microscopy (Cell imaging platform of the IBS) of Gag-mCherry, GFP-p40Phox, CHMP-NS3-blue, and CHMP-NS3-green was performed using an Olympus IX81 inverted microscope equipped with a 60× NA1.42 objective (PlanAPON60× Olympus, Hamburg, Germany), and CSU-X1 confocal head (Yokogawa, Tokyo, Japan). Excitation lasers source (iLaunch, GATACA system, Massy, France) was used for excitation at the suitable wavelengths and power settings. Images were collected employing the Metamorph software 7.10 (Molecular Devices, San Jose, CA, USA), via the adapted emission filters set, using a 16b/pixel 512 × 512 EMCCD (iXon Ultra, Andor, Belfast, Northen Ireland).

TIRF video-microscopy was conducted using an inverted microscope (iMIC 2.0, Till-Photonics, Planegg, Germany) equipped with an alpha-Plan-Apochromat 63x/1.46 objective lens (Zeiss, Jena, Germany). Image acquisition was performed with an iXon U897 EMCCD camera (Andor, Belfast, Northen Ireland). Cells were maintained at 37 °C in a 5% CO_2_ environment in a controlled chamber (Biovalley, Nanterre, France). Time-lapse movies were recorded over a 10 min duration, with images taken at intervals of 450 ms.

Electron microscopy: Ultrathin sections were observed using an FEI G2 Tecnai transmission electron microscope (TEM) equipped with an Orius SC1000 CCD camera.

### 2.4. Single Particle Tracking

The open-source Icy software 2.0.3 (https://icy.bioimageanalysis.org/, accessed on 16 October 2020) was used to semiautomatically track a large number of individual Gag-mCherry puncta [69]. The procedure included defining the region of interest at the cell’s leading edge, where Gag-mCherry spots were not densely distributed. Gag-mCherry particles were detected with the spot detector plug-in, and their tracks trajectories were subsequently determined using the spot tracking plug-in [69,70,71]. Manually, tracks at the edge of the region of interest and short tracks (≤2) were excluded. Subsequently, the track manager tool was employed to calculate velocities and tracking durations for each particle. Statistical analysis was conducted using GraphPad Prism 9 software.

## 3. Results

### 3.1. Expression and Autocleavage of CHMP-NS3-FP

Immunoblotting experiments were conducted to assess protein expression of CHMP-NS3-FP constructs in HEK-293 cells. Control cells treated with DMSO exhibited a time-dependent accumulation of cleaved NS3-FP proteins after transfection with CHMP2A-NS3-FP (Figure 2a), CHMP3-NS3-FP (Figure 2b, left panels), and CHMP4B-NS3-FP (Figure 2c). The raw dataset for Western blots is available in Appendix A. Glecaprevir treatment led to the temporal accumulation of noncleaved full-length CHMP-NS3-FP proteins as early as one hour (Figure 2a–c, right panels), and longer incubation times led to an increased presence of uncleaved CHMP-NS3-FP proteins. Notably, CHMP4B-NS3-FP showed the highest accumulation of uncleaved fusion proteins upon Glecaprevir treatment, while CHMP2A-NS3-FP reached its maximum after 4 h, and CHMP3-NS3-FP was least efficient in the inhibition of autocleavage (Figure 2a–c).

In order to assess the stability of the CHMP-NS3-FP proteins over time, a pulse-chase experiment was conducted. Notably, an increase in the amount of NS3-FP protein was observed, suggesting that newly expressed CHMP2A-NS3-FP, CHMP3-NS3-FP, and CHMP4B-NS3-FP proteins are autocleaved (Figure 2d–f). Remarkably, the quantity of full-length CHMP2A-NS3-FP, CHMP3-NS3-FP, and CHMP4B-NS3-FP was not equal and corresponded to the efficiency of fusion protein generation upon Glecaprevir treatment.

We conclude that the addition of Glecaprevir leads to a rapid and stable accumulation of full-length CHMP-NS3-FP proteins over time.

### 3.2. Cellular Localization of CHMP-NS3-FP Proteins

We next analyzed the localization of transiently expressed CHMP-NS3-FP proteins in HeLa CCL2 cells. In the DMSO group, transfection with CHMP-NS3-FP constructs resulted in a diffuse staining pattern, corresponding to the expression of cleaved NS3-FP (Figure 3a–c). In contrast, after a 4 h Glecaprevir treatment, while CHMP3-NS3-FP exhibited a diffuse cytosolic staining with few puncta at perinuclear sites and plasma membrane, and CHMP4B-NS3-blue and CHMP2A-NS3-blue showed some puncta at the plasma membrane and a more pronounced accumulation at perinuclear sites (Figure 3a). To determine the nature of the perinuclear staining, we cotransfected the CHMPs-NS3-blue constructs with the GFP-p40Phox plasmid, which recognizes PtdIns(3)P-enriched early endosomes [65]. We observed a partial colocalization of uncleaved CHMP2A-NS3-blue, CHMP3-NS3-blue or CHMP4B-NS3-blue proteins, and GFP-p40Phox following Glecaprevir treatment (Figure 3b, right panels).

To determine the localization of CHMP-NS3-FP proteins in the context of HIV-1 Gag VLP budding, we cotransfected the CHMP-NS3-FP constructs with Rev-independent HIV-1 Gag (hereafter called Gag) [63]. This showed that DMSO treated cells displayed cleaved NS3-FP proteins distributed throughout the cytosol. Upon Glecaprevir treatment, CHMP2A-NS3-green, CHMP3-NS3-green, and CHMP4B-NS3-green proteins colocalized with Gag-mCherry at the plasma membrane (Figure 3c, right panels). An average of 40% and 60% of Gag-mCherry spots colocalized with CHMP4B-NS3-green and CHMP2A-NS3-green, respectively (Figure 3d).

We conclude that Glecaprevir administration permits imaging of Gag colocalization with CHMP2A-NS3-green, CHMP3-NS3-green, and CHMP4B-NS3-green at the plasma membrane, indicative of a prolonged half-life of uncleaved CHMP-NS3-FP proteins at HIV-1 Gag budding sites.

### 3.3. Partial Inhibition of Gag VLP Release by CHMP-NS3-FP Proteins

We evaluated Gag in released VLPs and in whole cell extract (WCE) using immunoblotting in HEK-293 cells (raw dataset for Western blots is available in Appendix A). As controls, we included uncleavable CHMP-mutNS3-FP proteins without NS3 protease cleavage site and catalytic inactive dominant negative GFP-VPS4A E228Q [14] (Appendix A). As expected, Gag cotransfection with GFP-VPS4A E228Q, CHMP2A-mutNS3-FP, or CHMP4B-mutNS3-FP resulted in a significant impairment of HIV-1 Gag VLP discharge, to the extent that emitted VLPs were undetectable compared to wild-type Gag VLP release (Figure 4a–c). We also tested the VPS4B R253A mutant that revealed slower kinetics in disassembling ESCRT-III CHMP2A-CHMP3 helical polymers in vitro [64] and demonstrated a reduction in VLP release (Figure 4f). In DMSO-treated cells, VLP emission was minimally affected when CHMP3-NS3-FP or CHMP4B-NS3-FP were expressed while CHMP2A-NS3-FP expression led to a 20% decrease in VLP release (Figure 4a–c).

Upon Glecaprevir treatment, uncleaved CHMP-NS3-FP proteins started to accumulate (Figure 4a–c, right panels), mimicking the effect of the mutation of the NS3 cleavage site in the CHMP-mutNS3-FP proteins (Figure 4a–c, right panels). Surprisingly, neither CHMP3-mutNS3-FP nor the presence of Glecaprevir in cells transfected with CHMP3-NS3-FP affected HIV-1 Gag VLP release (Figure 4c). However, consistent with previous findings, CHMP3-YFP completely impaired HIV-1 budding [17] (Figure 4d). Although CHMP3 is not strictly required for HIV-1 budding [52], it synergizes with CHMP2A to enhance HIV-1 budding efficiency [53]. Accordingly, we observed a synergistic effect of uncleaved CHMP3-NS3-FP and CHMP2A-NS3-FP on Gag VLP release (Figure 4e). Importantly, expression of CHMP2A-NS3-FP consistently reduced VLP liberation by ~40%. However, the most pronounced reduction in VLP discharge was observed in cells expressing CHMP4B-NS3-FP, resulting in a decrease in more than 78% without affecting intracellular Gag protein levels (Figure 4a,b,f).

### 3.4. Cell-Line-Specific and Dose-Dependent Effects of CHMP-NS3-FP Proteins

Our assay was designed as an imaging tool for studying ESCRT-III proteins, prompting us to assess its efficacy in the HeLa CCL2 cell line. Notably, unlike Hek293 cells, HeLa CCL2 cells have been documented to express BST2 tetherin proteins, known for their ability to restrict the release of HIV-1 virions [72,73]. We thus include a HeLa Kyoto cell line which has been BST2-deleted and stably expresses MKLP1-GFP [66]. As previously reported, HeLa Kyoto BST2+ cells, when transfected with Gag, did not liberate VLPs, while HeLa Kyoto BST2- cells did (Appendix A) [73]. Surprisingly, our HeLa CCL2 cells displayed a significant accumulation of free VLPs, indicating a moderate expression of BST2 in this particular HeLa CCL2 cell line. In support of this, immunoblotting revealed reduced BST2 expression in HeLa CCL2 compared to HeLa Kyoto BST2+ cells, with no detectable expression of BST2 in either Hek293 or HeLa Kyoto BST2- cells (Appendix A). Furthermore, cotransfection with Vpu, the BST2 inhibiting factor, poorly enhanced the accumulation of free VLPs in HeLa CCL2 cells (Appendix A). In conclusion, our results suggest that the weak expression of BST2 in the HeLa CCL2 cell line does not efficiently impede the liberation of VLPs.

To assess assay efficiency in HeLa Kyoto BST2− and Hek293 cell lines, we cotransfected these cells with Gag and varying amounts of CHMP-mutNS3-green or Glecaprevir-treated CHMP-NS3-green constructs, as outlined in Figure 5 (raw dataset for Western blots is available in Appendix A). It is worth noting that in HeLa Kyoto BST2− cells, exposure to Glecaprevir led to a modest reduction in VLP release when using the CHMP4B-NS3mut-green and CHMP2A-NS3mut-green constructs. Nonetheless, the results unequivocally demonstrate a dose-dependent decrease in HIV-1 VLP release (Figure 5). We conclude that budding inhibition efficiency by Glecaprevir-treated CHMP4B-NS3-green and CHMP2A-NS3-green proteins is cell-line-dependent and correlates with expression.

### 3.5. Quantitative Analysis of HIV-1 Gag VLP Dynamics Reveals Drug-Dependent Prolongation of Gag VLP Retention

We hypothesized that CHMP-NS3-FP’s inhibition of VLP liberation could potentially lead to a delay in VLP release. To precisely measure the timing of VLP release at the cellular level, we conducted thorough TIRF video-microscopy imaging on HeLa CCL2 and HeLa Kyoto BST2- cells. We recorded movies spanning 10 min with a temporal resolution of 0.45 s per frame. It is important to mention that HeLa Kyoto BST2- cells cotransfected with Gag/Gag-mCherry/CHMP2A-NS3-FP or CHMP4B-NS3-FP and treated with Glecaprevir exhibited a rounded morphology in over 25% of the cells, indicating a potential toxic effect associated with the transfected constructs. Interestingly, this phenomenon was not observed in HeLa CCL2 cells. We tracked Gag-mCherry spots and conducted quantitative analyses of their maximum velocity and duration (details in Materials and Methods). Notably, the distribution of these values appeared consistent across different cells within each experimental condition (Appendix A). Gag-mCherry spots might correspond to (i) VLPs in the process of assembly or assembled but not yet released, (ii) VLPs release from the plasma membrane but tethered by the glycocalix and/or proteins or adhering to the glass surface, and (iii) free VLPs moving in the narrow space between the cell and the coverslip. Free-moving virions have been described to exhibit a high maximum velocity [56,74]. Consistent with previous findings [56], we observed in both HeLa CCL2 and HeLa Kyoto BST2- cells that the coexpression of VPS4A_E228Q significantly reduced the population of spots with high maximum velocity (>0.625 μm/s) (Appendix A). Furthermore, the expression of both CHMP2A-NS3-FP and CHMP4B-NS3-FP led to a reduction in the number of spots with high maximum velocity (Appendix A). We conclude that in HeLa CCL2 cells, both CHMP2A-NS3-FP and CHMP4B-NS3-FP proteins markedly reduce the number of VLPs released at the individual cell level.

In addition, these findings affirm the accuracy of our tracking analysis.

Next, we analyzed the individual tracking time duration of Gag-mCherry dots. As expected, cells transfected with a mixture of Gag-mCherry/Gag and Vps4A_E228Q displayed a strong increase in spots tracked for the whole movie duration (19.5 ± 5.8% compared to 1.5 ± 0.9% in control, *p* < 0.0001, unpaired *t*-test; Figure 6a). These spots most probably correspond to VLPs assembled but blocked in the process of membrane fission. For CHMP-NS3-FP constructs, Glecaprevir addition enhanced the duration time in a specific way. CHMP2A-NS3-FP triggered a 3.4-fold enhancement of Gag-mCherry dots that lasted for the entire 10 min recording time from 1.4 ± 0.7% in DMSO to 4.1 ± 1.3% in the Glecaprevir group (*p* = 0.0072, unpaired *t*-test; Figure 6c). In contrast, CHMP4B-NS3-FP enhanced the proportion of tracks that had a duration greater than one minute from 7.7 ± 0.5% in DMSO to 18.7 ± 2.8% in the Glecaprevir group (*p* < 0.0001, unpaired *t*-test; Figure 6d). Notably, the observed phenotype of CHMP2A-NS3-FP and CHMP4B-NS3-FP fusion enhancing tracks lasting time was significantly improved when examining colocalized spots (Figure 6c,d).

Our findings underscore a drug-dependent prolongation of the lifetime of HIV-1 Gag VLPs at a cell’s surface. This effect was particularly pronounced at sites where CHMP4B-NS3-FP and CHMP2A-NS3-FP accumulate.

### 3.6. Electron Microscopy Reveals Drug-Induced Impairment of Late-Stage HIV-1 VLP Budding

To gain insight into the structure of VLPs within cells, both in the presence and absence of the drug, we conducted electron microscopy imaging. Specifically, we examined 293FS cells that had been cotransfected with CHMP2A-NS3 or CHMP4B-NS3 along with Gag. These cells were subjected to high-pressure freezing, resin embedding, and subsequent imaging.

In cells treated with DMSO, we observed only a minimal presence of cell-associated VLPs (as depicted in Figure 7a,g). In stark contrast, cells treated with Boceprevir, an NS3 inhibitor, for a duration of 2 h exhibited numerous particles that remained tethered to the parental cell via membrane stalks (Figure 7). This observation suggests that VLPs had assembled but were unable to undergo release.

In conclusion, our findings suggest that fusion proteins CHMP2A-NS3 or CHMP4B-NS3 impair the late stage of budding, specifically impeding plasma membrane fission.

## 4. Discussion

A large number of cellular membrane remodeling processes are catalyzed by the ESCRT machinery, which can be efficiently blocked by dominant negative forms of ESCRT-III or VPS4 [22,75,76,77]. However, strong dominant negative inhibition affects multiple cellular processes, and the observed effect on a given process may be influenced by blocking many essential cellular functions, not only the targeted pathway. To address this issue, we developed an inducible and transient ESCRT-III inhibition system based on the NS3 hepatitis C protease, described by the Tsien lab [78,79]. In brief, a cis-acting NS3 protease was fused to the C-terminus of ESCRT-III via a short linker containing the NS3 cleavage site. To track the fusion protein, a fluorescent protein (FP) was further genetically fused C-terminally of NS3. The CHMP-NS3-FP fusion proteins underwent autocleavage, producing transiently wild-type-like CHMP, consistent with previous reports employing cis-acting NS3 [78]. NS3-catalyzed autocleavage likely occurs immediately upon expression since no fusion protein can be detected upon one hour of expression. Likewise, inhibition of autocleavage with Glecaprevir led to the transient accumulation of full-length CHMP-NS3-FP fusion proteins detectable within just 30 min, and maximum levels after 4 h of treatment for CHMP2A-NS3-green and CHMP3-NS3-green, or 24 h for CHMP4B-NS3-Green.

C-terminal fusions of CHMP1 have been shown to exert a dominant negative effect on vesicle trafficking [80], and HIV-1 budding can be inhibited with CHMP3-YFP, CHMP3-RFP, and CHMP4A/C-RFP fusion proteins [17,18]. Furthermore, N-terminal ESCRT-III fusion proteins were also reported to be dominant negative [18,81]. We chose to target the C-terminus because N-terminal fusions of CHMP proteins act differently from C-terminal ESCRT-III fusions. N-terminal fusions likely interfere with membrane interaction due to the N-terminal membrane insertion domains present in most ESCRT-III proteins, which may in turn also affect polymerization on membranes [34,43,82]. In contrast, the effect of C-terminal fusions is likely linked to the function of VPS4, which is required for ESCRT-III remodeling and disassembly [25]. Here, we showed that uncleavable CHMP-NS3-FP fusion proteins containing CHMP2A and CHMP4 effectively impede the release of HIV-1 Gag VLPs. Notably, CHMP3 demonstrates a limited impact on inhibiting VLP discharge, in contrast to the strong inhibitory effects observed with CHMP3-YFP and CHMP3-RFP [17,18]. We hypothesize that the observed dominant negative effect of C-terminal fusions is due to the overall conformation of the fusion protein, which can sterically interfere with the activity of VPS4. This is consistent with CHMP2B, CHMP3, and CHMP4B tagged with GFP via a long flexible linker (LAP-tag) showing no detectable perturbation of abscission in HeLa cells [37]. We therefore propose that CHMP4B-NS3-FP and CHMP2A-NS3-FP perturb VPS4 activity, which in turn reduces VLP release. In contrast, CHMP3-NS3-FP does not affect VPS4 activity, suggesting that its structure is compatible with VPS4 function. Blocking autocleavage of CHMP4B and CHMP2A by NS3 with Glecaprevir does not fully inhibit VLP release, and, notably, blocking autocleavage of CHMP3-NS3-FP has no effect on VLP release, but enhances the effect of CHMP2A-NS3-FP, in agreement with the reported synergy of CHMP2A-CHMP3 on HIV-1 budding [53] and CHM2A-CHMP3 heteropolymer formation in vitro [34]. Interestingly, a VPS4B mutant that was previously shown to slow down the kinetics of ESCRT-III disassembly in vitro [64] showed a similar reduction in VLP release as inhibition of CHMP4B and CHMP2A autocleavage. We propose that the inhibited fusion proteins slow down the ESCRT-III/VPS4 machinery, leading to a reduced detection of VLPs.

The effect of Glecaprevir, the accumulation of full-length CHMP-NS3-FP proteins, appears to be counterbalanced by protein synthesis, resulting in the continuous presence of cleavage products, NS3-FP and CHMP proteins. The latter are supposed to cooperate with native CHMPs in regular ESCRT-III function. Strikingly, the full-length CHMP-NS3-FP protein accumulation correlated with their dominant negative activity, suggesting that the dosage of full-length CHMP-NS3-FP proteins determines the slow-down effect of ESCRT-III function.

We demonstrated that the reduction in VLP release is cell-line-dependent. Hek293 cells exhibited a stronger dominant negative activity of CHMP-NS3-FP, whereas HeLa cells displayed lower inhibition of VLP release. Notably, HeLa cells have been reported to consistently express BST2, which traps assembled HIV-1 particles at the cell surface and is countered by the HIV-1 accessory protein Vpu [72,73]. In this study, we observed that BST2 expression is higher in HeLa Kyoto cells compared to HeLa CCL2 cells. Interestingly, in HeLa CCL2 cells, HIV-1 VLP particles were efficiently liberated even in the absence of Vpu, although its presence did enhance the release slightly. These findings are consistent with a certain BST2 threshold concentration being necessary for sequestering HIV-1 particles.

We further showed that the NS3-FP fusions affect ESCRT-III localization. Full-length CHMP3-NS3-green was mainly distributed throughout the cytosol, while CHMP2A-NS3-FP and CHMP4B-NS3-FP revealed some punctuate staining at the plasma membrane and at endosomes and/or multivesicular endosomes. HIV-1 Gag has been shown to accumulate at the plasma membrane until Gag VLP bud formation, which triggers the recruitment of ESCRT-III followed by Vps4A [54,55,59]. This recruitment process is of short duration, typically lasting less than two minutes, resulting in a relatively low fraction of HIV-1 budding sites displaying colocalization with ESCRT complexes, estimated to be between 1.5% and 3.4% [57,58].

In our current study, we provided compelling evidence that noncleaved CHMP4B-NS3-FP and CHMP2A-NS3-FP proteins exhibit an extended half-life at HIV-1 budding sites, concurrently demonstrating a substantial colocalization of these proteins with Gag-mCherry. In addition, expression of CHMP4B-NS3-FP or CHMP2A-NS3-FP proteins extends the period during which VLP particles could be tracked at the cell surface. We also directly observed an increased number of particles connected to the parental cell by a plasma membrane neck. From these observations, we infer that the accumulation of CHMP4B-NS3-FP and CHMP2A-NS3-FP proteins acts to decelerate the process of plasma membrane fission at HIV-1 budding sites.

In summary, we set up a novel inducible ESCRT-III system that will be helpful for imaging ESCRT-III at HIV-1 budding sites to obtain more insight into its structure and function. Finally, the system will be useful to study the large plethora of ESCRT-dependent membrane remodeling processes.

## Figures and Tables

**Figure 1 viruses-15-02289-f001:**
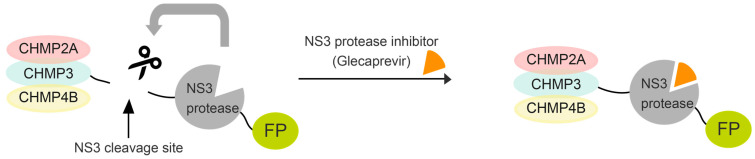
Schematic illustrating our method for transient expression of ESCRT-III fused to NS3 protease and a fluorescent protein using a drug. ESCRT-III proteins are fused to the NS3 cleavage site, NS3 protease, and fluorescent protein. Cells transfected with this construct express wild-type ESCRT-III proteins. Upon addition of the drug, a full-length fusion protein accumulates. For imaging and immunoblotting detection, fluorescent proteins (mostly mNeonGreen) and a Flag tag were further added to the NS3 C-terminus, creating CHMP4B/2A/3-NS3-FP-Flag constructs (hereafter collectively referred to as CHMP-NS3-FP).

**Figure 2 viruses-15-02289-f002:**
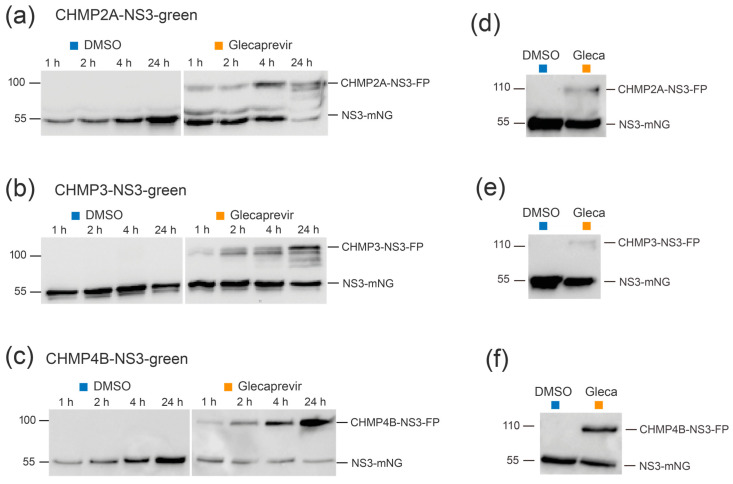
Complete CHMP-NS3-FP proteins are stably expressed over time. (**a**–**c**) Representative immunoblot experiments were performed using whole cell extracts from HEK293 cells transfected with CHMP2A-NS3-green (2 μg) (**a**), CHMP3-NS3-green (2 μg) (**b**), or CHMP4B-NS3-green (1 μg) (**c**) and treated with DMSO or Glecaprevir for the indicated duration. (**d**–**f**) Pulse-chase experiment. HEK293 cells transfected with CHMP2A-NS3-green (**d**), CHMP3-NS3-green (**e**), or CHMP4B-NS3-green (**f**) were treated with DMSO or Glecaprevir for 4 h. Subsequently, the medium was washed away and replaced with fresh medium, and whole cell extracts were obtained 24 h later.

**Figure 3 viruses-15-02289-f003:**
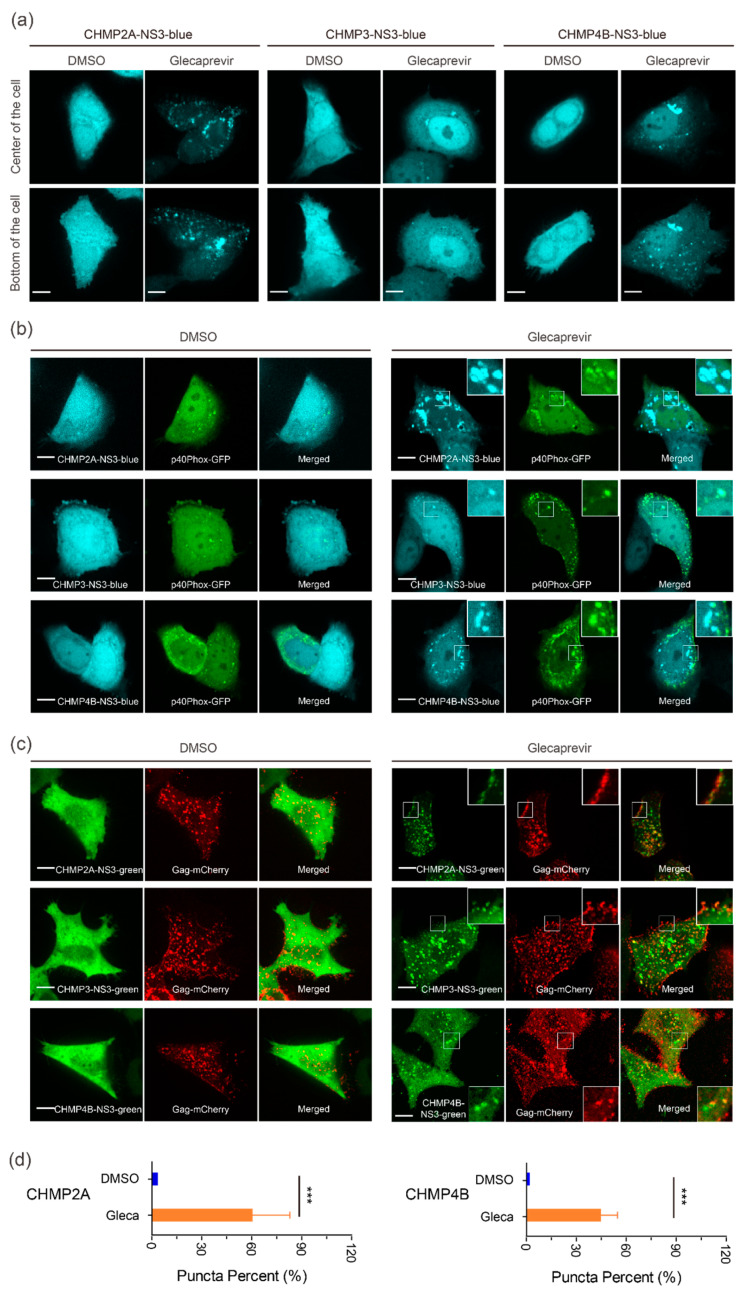
In vivo localization of CHMPs-NS3-FP proteins. (**a**) Distribution on CHMP-NS3-FP in HeLA CCL2 cells: Cells transfected with CHMP2A-NS3-blue (left panels), CHMP3-NS3-blue (middle panels), or CHMP4B-NS3-blue (right panels) were treated for 4 h with DMSO or Glecaprevir as indicated. (**b**) CHMPs-NS3-blue accumulate in endosomes: Cells cotransfected with GFP-p40Phox and CHMP2A-NS3-blue (upper panels), CHMP3-NS3-blue (middle panels), or CHMP4B-NS3-blue (lower panels) were treated for 4 h with DMSO or Glecaprevir as indicated. (**c**) Colocalization of CHMP-NS3-green proteins with Gag-mCherry: Cells cotransfected with Gag, Gag-mCherry and CHMP2A-NS3-green (upper panels), CHMP3-NS3-green (middle panels), or CHMP4B-NS3-green (lower panels) were treated for 4 h with DMSO or Glecaprevir as indicated. Scale bars are 10 μm. (**d**) Quantification of Gag-mCherry spots as a proportion colocalizing with CHMP2A-NS3-green or CHMP4B-NS3-green (mean ± SD, n > 230 spots for each condition from 3 cells). The statistical significance (***, indicating *p* < 0.001), was determined through a Mann–Whitney test.

**Figure 4 viruses-15-02289-f004:**
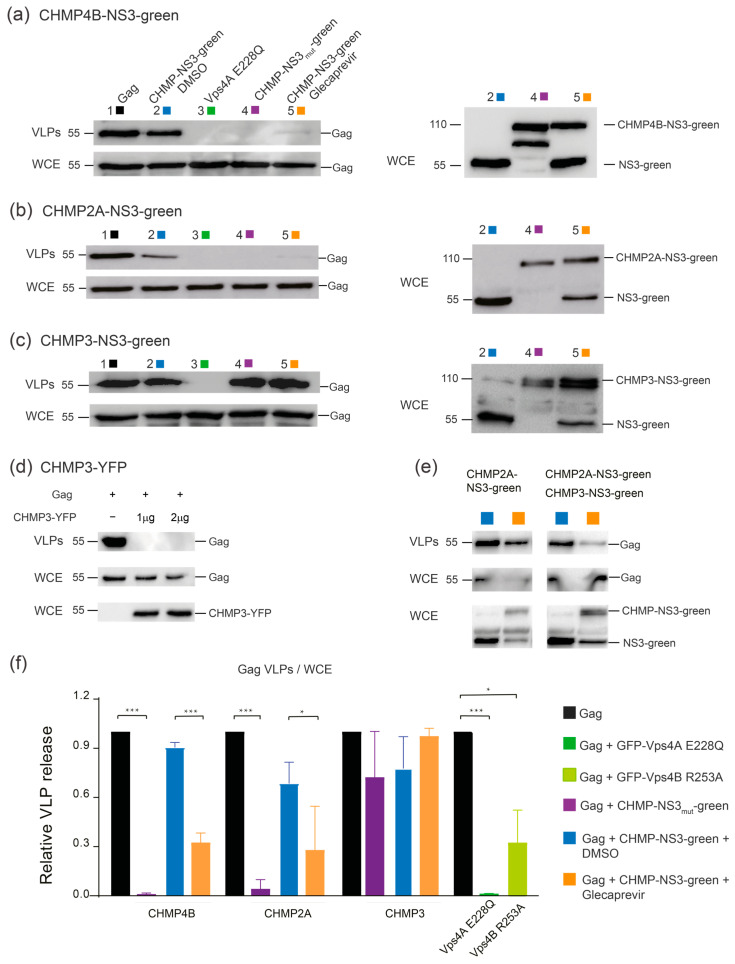
Inhibition of VLP release by CHMP4B-NS3-green, CHMP2A-NS3-green, and CHMP3- NS3-green: immunoblot analysis. (**a**–**c**) Inhibition of VLP release in cells cotransfected by Gag (0.5 μg), GFP-Vps4A E228Q (1 μg), and CHMP4B-NS3-green (1 μg) (**a**), CHMP2A-NS3-green (4 μg) (**b**), and CHMP3-NS3-green (4 μg) (**c**). Representative immunoblot experiment depicting the following: Left panels (i) upper line: HIV-1 VLP pellet, (ii) second line: Gag HIV-1 cellular expression (WCE: whole cell extract), and right panels: CHMPs-NS3-green cellular expression. Hek293 cells were transfected with the following: column 1: Gag, column 2: Gag and CHMPs-NS3-green treated with DMSO for 2 h, column 3: Gag and GFP-Vps4A E228Q, column 4: Gag and CHMPs-mut-NS3-green, column 5: Gag and CHMPs-NS3-green treated with Glecaprevir for 4 h. (**d**) Representative immunoblot depicting Hek293 cells transfected with Gag and CHMP3-YFP. The upper lines represent HIV-1 VLPs released, the second lines indicate HIV-1 cellular expression (WCE: whole cell extract), and the third lines depict CHMP3-YFP cellular expression. (**e**) Synergy between CHMP3-NS3-green and CHMP2A-NS3-green. To enhance sensitivity, the amount of transfected CHMP2A-NS3-green was reduced to 2 μg, allowing for a slight impairment of VLP release. Cotransfection of Gag (0.5 μg), CHMP2A-NS3-green (2 μg), and CHMP3-NS3-green (2 μg) clearly enhance the VLP release inhibition. Representative immunoblot depicting Hek293 cells transfected with left panels: Gag and CHMP2A-NS3-green, right panels: Gag, CHMP2A-NS3-green, and CHMP3-NS3-green, treated with DMSO or Glecaprevir as indicated. The upper lines represent released HIV-1 VLPs, the second lines indicate HIV-1 cellular expression, and the third lines depict CHMPs-NS3-green cellular expression. (**f**) VLP release of the tested constructs analyzed by Western blot. Data are presented for Gag, GFP-Vps4A E228Q, GFP-Vps4B R253A, CHMP4B-NS3-green, CHMP2A-NS3-green, and CHMP3-NS3-green, as indicated (mean ± SD, n = 3 for each condition). The statistical significance (* for *p* < 0.05 and *** for *p* < 0.001), were determined through a Mann–Whitney test. HIV-1 Gag proteins were detected using anti-p24 antibody, CHMP3-YFP were detected using anti-GFP antibody, while CHMPs-NS3-green were detected using Anti-Flag antibody.

**Figure 5 viruses-15-02289-f005:**
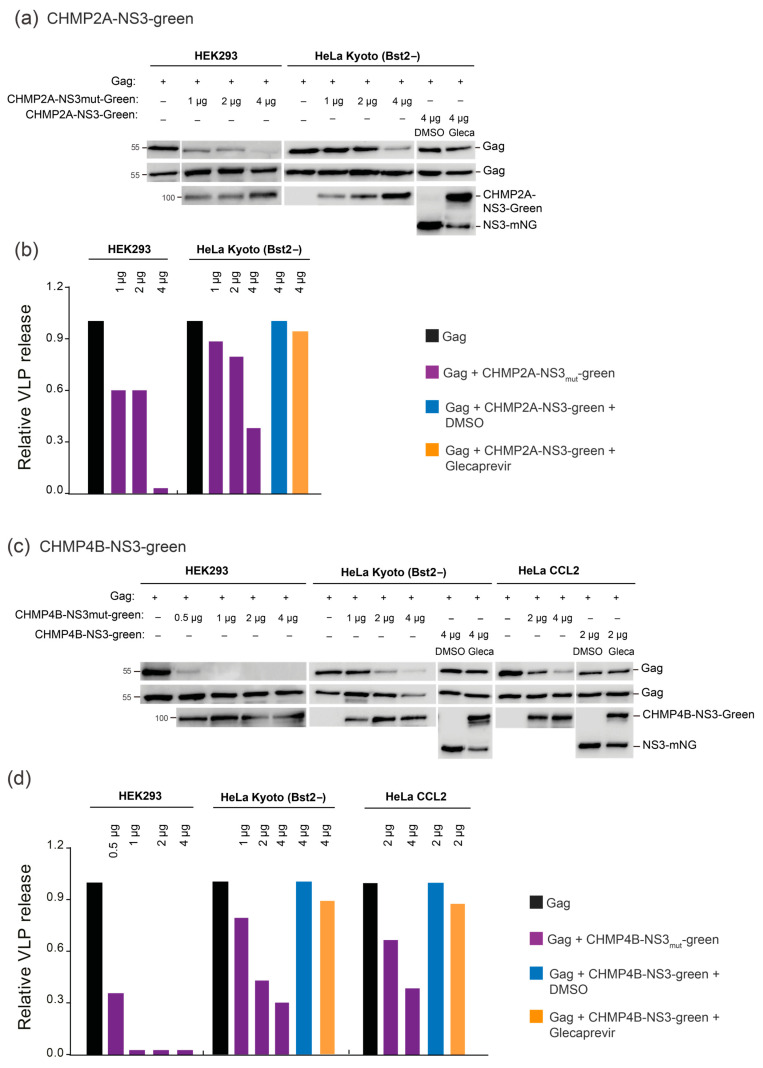
Dose- and cell-line-dependent analysis via immunoblotting. (**a**,**b**) Inhibition of VLP release in cells cotransfected with Gag (0.5 μg) alongside varying quantities of either CHMP2A-NS3mut-green or CHMP2A-NS3-green, treated with DMSO or Glecaprevir. (**a**) Illustrative immunoblot experiment showing (top line) HIV-1 VLP pellet, (second line) Gag HIV-1 cellular expression (WCE: whole cell extract), and (lower line) CHMP2A-NS3-green cellular expression. Hek293 cells or HeLa Kyoto (Bst2−) were transfected as indicated. (**b**) Quantitative analysis of the assessed constructs via Western blot. (**c**,**d**) Inhibition of VLP release in cells cotransfected with Gag (0.5 μg) and varied quantities of either CHMP4B-NS3mut-green or CHMP4B-NS3-green, treated with DMSO or Glecaprevir. (**c**) Representative immunoblot experiment exhibiting (top line) HIV-1 VLP pellet, (second line) Gag HIV-1 cellular expression (WCE: whole cell extract), and (lower line) CHMP4B-NS3-green cellular expression. Hek293 cells, HeLa Kyoto (Bst2−), or HeLa CCL2 were transfected as indicated. (**d**) Quantitative assessment of the tested constructs via Western blot. HIV-1 Gag proteins were detected using an anti-p24 antibody, while CHMPs-NS3-green were detected using an Anti-Flag antibody. Note that the exposure time for Western blot revelation was typically 30 s for Hek293 cells and 5 min for HeLa CCL2 and HeLa Kyoto cells.

**Figure 6 viruses-15-02289-f006:**
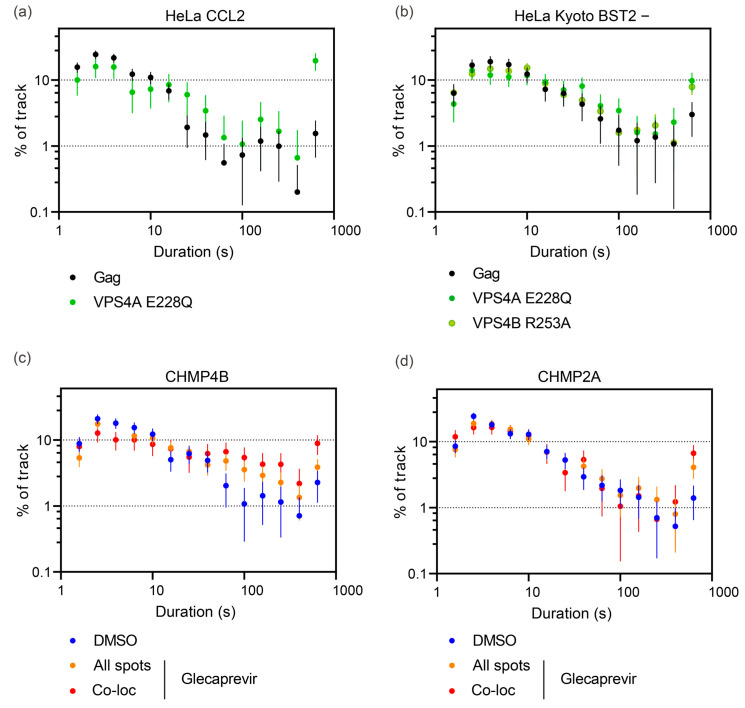
Tracking time duration of Gag-mCherry spots. (**a**,**c**,**d**) Frequency distributions of spot tracking time duration in individual HeLa CCL2 cells. The cells were transfected by Gag/Gag-mCherry alone or Gag/Gag-mCherry along with Vps4A E228Q (**a**), CHMP4B-NS3-green (**c**), and CHMP2A-NS3-green (**d**), and then treated or not by Glecaprevir, as indicated. The error bars represent the standard deviation (n = 649; 231; 1874; 1669; 1142; 1527 spots from 4; 5; 8; 8; 8; 8 cells for Gag/Gag-mCherry, along with Vps4A E228Q, CHMP2A-NS3-green DMSO, CHMP2A-NS3-green Glecaprevir, CHMP4B-NS3-green DMSO, and CHMP4B-NS3-green Glecaprevir, respectively). (**b**) Frequency distribution of spot tracking time in individual HeLa Kyoto Bst2- cells. These cells were transfected with Gag/Gag-mCherry alone, Gag/Gag-mCherry with Vps4A E228Q, or Gag/Gag-mCherry with Vps4B R253A. The error bars represent the standard deviation (n = 883; 424; 1744 spots from 9; 6; 9 cells for Gag/Gag-mCherry along with Vps4A E228Q and Vps4B R253A, respectively).

**Figure 7 viruses-15-02289-f007:**
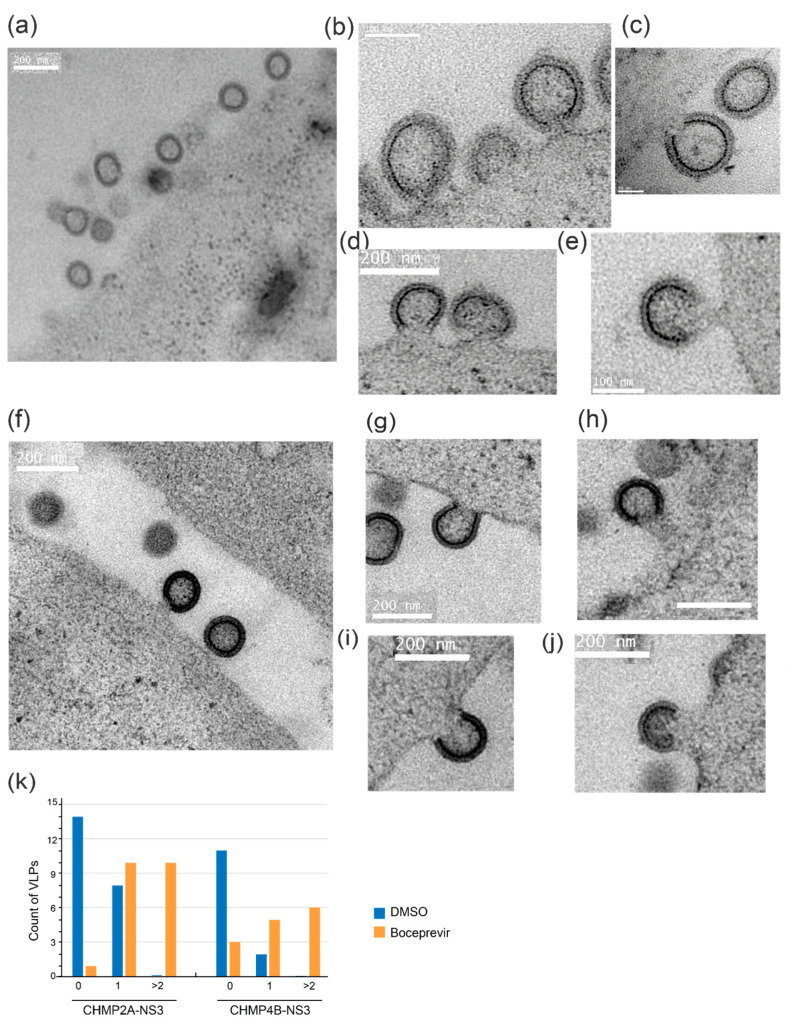
Electron microscopy images of 293FS cells showing VLP arrested in budding at the plasma membrane. (**a**–**e**) 293FS cells transfected with Gag and CHMP2A-NS3 and treated for 2 h with either DMSO (**a**) or Boceprevir (**b**–**e**). (**f**–**j**) 293FS cells transfected with Gag and CHMP4B-NS3 and treated for 2 h with either DMSO (**f**) or Boceprevir (**g**–**j**). EM images depict NS3 inhibitor action in arresting VLPs budding at the plasma membrane. Scale bars indicate 50 nm (**c**), 100 nm (**b**,**e**), and 200 nm (**a**,**d**,**f**–**j**). (**k**) Quantification of VLPs connected to the membrane per field of view in 293FS cells transfected with either CHMP4B-NS3 or CHMP2A-NS3 and subjected to treatment with DMSO or Glecaprevir as indicated (cell count is n = 3 for each condition).

## Data Availability

Spot tracking datasets are accessible at https://doi.org/10.57745/69UNAM.

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
