# Peer review of "An Inducible ESCRT-III Inhibition Tool to Control HIV-1 Budding"

_viruses, 2023, doi:10.3390/v15122289_

Round 1

Reviewer 1 Report

Comments and Suggestions for Authors

This manuscript reports the development of a clever new tool to transiently inhibit the activity of ESCRT-III. Briefly, the authors fuse several ESCRT-III subunits to a viral protease to generate auto-cleavable proteins that can be stabilized by treatment with a protease inhibitor, causing a dominant negative effect and interfering with their function. As proof of principle, the authors use this system to extend the lifetime of membrane associated Gag based VLPs. Altogether, these findings will be of interest not only in the field of virology but also in the broader field of cell biology. I consider this paper to be suitable for Viruses, subject to some improvements in the data, which is somewhat untidy in places. Specifically:

1) Whilst the authors show that the CHMP2A and CHMP4B-based constructs accumulate upon addition of Glecaprevir, co-localize with endosomal markers and membrane-bound Gag and inhibit VLP production (Figures 1-3), I don’t think that these claims can be extended to the CHMP3-based construct, which overall is not functional. I suggest removing the data generated with this construct from the manuscript to simplify the message and for quality consistency.

2) The co-localization of CHMP4B-NS3-FP with Gag in the images shown in Figure 2 is not obvious to me – For each sequence of microscopy images, corresponding fluorescent intensity plots could be shown on the right for further demonstration of the co-localization.

3) The data shown to demonstrate cell line-specific and dose-dependent effects of the fusion proteins is confusing and needs tidying up to be convincing. Rather than showing the current table, where there is no pattern of consistent conditions tested for all constructs, the authors should choose one pair of representative CHM-NS3-FP and CHMP-mutNS3-FP and three consistent concentrations (e.g., 0.5, 2 & 4 ug) and show the graphs and western blots for all three cell lines in a similar way to what they’ve done in Figure 3 so that data can be easily compared. The rest of constructs can be moved to the supplementary material as a table like the one they have now in the main text. 

4) Figure 5 would be more compelling by adding a graph quantifying how many VLPs bound to membrane were found in each condition per field of view. Why do the authors use Boceprevir instead of Glecaprevir (used throughout the manuscript) for the EM studies?

Minor typos:

Line 368-369: (d) is showing CHMP2A-NS3-green (not CHMp4B)

Lines 372 and 376: “respectfully” needs to be replaced by “respectively”

Author Response

We would like to thank the reviewer for taking the time to review this manuscript. Please find the detailed responses below and the corresponding revisions/corrections highlighted in red track changes in the re-submitted files.

Comments and Suggestions for Authors

This manuscript reports the development of a clever new tool to transiently inhibit the activity of ESCRT-III. Briefly, the authors fuse several ESCRT-III subunits to a viral protease to generate auto-cleavable proteins that can be stabilized by treatment with a protease inhibitor, causing a dominant negative effect and interfering with their function. As proof of principle, the authors use this system to extend the lifetime of membrane associated Gag based VLPs. Altogether, these findings will be of interest not only in the field of virology but also in the broader field of cell biology. I consider this paper to be suitable for Viruses, subject to some improvements in the data, which is somewhat untidy in places. Specifically:

1) Whilst the authors show that the CHMP2A and CHMP4B-based constructs accumulate upon addition of Glecaprevir, co-localize with endosomal markers and membrane-bound Gag and inhibit VLP production (Figures 1-3), I don’t think that these claims can be extended to the CHMP3-based construct, which overall is not functional. I suggest removing the data generated with this construct from the manuscript to simplify the message and for quality consistency.

We have revised our manuscript to explicitly specify that only CHMP2A-NS3-FP and CHMP4B-NS3-FP exhibit inhibitory properties on VLP release. As a result, we have made adjustments to the abstract.

However, we respectfully disagree with the notion that the CHMP3-based construct is non-functional. In fact, the CHMP3-NS3-FP construct, retains the ability to co-localize with membrane-bound Gag-mcherry and synergizes with the CHMP2A-based construct in inhibiting VLP release demonstrating that this protein is properly folded and active.

2) The co-localization of CHMP4B-NS3-FP with Gag in the images shown in Figure 2 is not obvious to me – For each sequence of microscopy images, corresponding fluorescent intensity plots could be shown on the right for further demonstration of the co-localization.

We have enhanced the quality of image showing CHMP4B-NS3-green co-localization with Gag-mcherry to fulfill reviewer comments.

3) The data shown to demonstrate cell line-specific and dose-dependent effects of the fusion proteins is confusing and needs tidying up to be convincing. Rather than showing the current table, where there is no pattern of consistent conditions tested for all constructs, the authors should choose one pair of representative CHM-NS3-FP and CHMP-mutNS3-FP and three consistent concentrations (e.g., 0.5, 2 & 4 ug) and show the graphs and western blots for all three cell lines in a similar way to what they’ve done in Figure 3 so that data can be easily compared. The rest of constructs can be moved to the supplementary material as a table like the one they have now in the main text. 

In response to the reviewer's requirements, we substituted Table 1 with a figure that includes Western blots and graphs. 

4) Figure 5 would be more compelling by adding a graph quantifying how many VLPs bound to membrane were found in each condition per field of view. Why do the authors use Boceprevir instead of Glecaprevir (used throughout the manuscript) for the EM studies?

We added a graph to address the reviewer's comment, illustrating the quantification of VLPs bound to the membrane.

Originally, Boceprevir was employed as the drug to inhibit NS3 activity in 293FS cells used for EM imaging. However, in experiments with Hek293 cells, we replaced Boceprevir with Glecaprevir due to its superior inhibitory properties.

Minor typos:

Line 368-369: (d) is showing CHMP2A-NS3-green (not CHMp4B)

Done

Lines 372 and 376: “respectfully” needs to be replaced by “respectively”

Done

Reviewer 2 Report

Comments and Suggestions for Authors

Reviewer comments on manuscript

Reference:       viruses-2696757

Authors:     Haiyan Wang, Benoit Gallet, Christine Moriscot, Mylène Pezet, Christine Chatellard, Jean-Philippe Kleman, Heinrich Göttlinger, Winfried Weissenhorn* and Cécile Boscheron*

Tittle:               An inducible ESCRT-III inhibition tool to control HIV-1 3 budding

In this research article, the team led by Weissenhorn and Boscheron describes a piece of research focused on the development of a bio-tool aimed at clarifying the mechanism of HIV-1 budding involving the host ESCRT-III machinery. Thus, self-cleavable hybrid species obtained by fusion of CHMP2A, /-3 or /-4B and hepatitis C virus NS3 protease are described, additionally labelled with fluorescent protein markers in order to monitor them. A key aspect of this approach lies on the inhibition of CHMP fusion protein self-cleavage through the concomitant use of an inhibitory agent (glecaprevir), leading to a modulable biological tool applicable for the visualisation of HIV-1 budding processes linked to ESCRT-III aimed at obtaining more detailed information, both functional and mechanistic, opening a promising line of research on the potential of this endogenous system for the as target for treatment of HIV-1 infection.

The paper is written in a direct, simple style. The main aspects related to the ESCRT-III system (and related to HIV-1 infection) are described in a rigorous and detailed fashion. After an introduction that sets the thematic context of the work, the material and methods section highlighted the techniques and protocols followed. The results section describes in detail the experiments carried out, including a rational justification of the data and further experiments proposed, which is highly welcome to follow properly the description of the data with the proper graphical support (Figures embedded in the core text and supplemental material available), summarized subsequently in an exhaustive discussion of all results previously depicted.

The manuscript is rationally and well organised and supported by experimental evidences. Obviously, this piece of work contains in some extent some follow-up aspects related to the previous work of the research group and others, but still maintains a high level of originality and novelty. The manuscript is adequately referred, supported by a vast, up-to-date bibliography (82 references). The material and methods are the usually used for these purposes and properly described.

After a critical reading, the present manuscript meets the minimum requirements expected, merging enough contents, enough interest and proper methodological support as to be considered for publication. In spite of what is said, there are still some minor objections for this reviewer exposed as follow:

1-. In the introduction section, page 2, the paragraph contained between lines 78 (“We find that…” ) and 83 (…Co-localizatio”) actually anticipates results of this work. Consequently, I would suggest to move forward to the end of the paper (i.e., page 16, lines 479 onwards).

2-. In parallel, in page 4, paragraph located between lines 158 (”to study the…) and 167 (…proteins (Figure 1a)” should be located before as anticipates the aim and objective of the research. Consequently, I would suggest to move forward to the beginning of the paper (i.e., page 2, lines78 onwards). Figure 1a would be split from the original Figure and relocated accordingly on page 2.

This change would help to understand the aim and objective of the research prior to the description of the results as currently deployed.

3-. Authors Would indicate the interest and projection of this study focused on ESCRT-III inhibition.

Is ESCRRT-III a validated target against HIV-1?

What is the projection of the work thinking in a close-time therapeutic application against HIV-1 infection?

Are the described fusion proteins clinically usable or just a research tool? 

Is ESCRT-III drugable?

What is the interest/projection of this study in terms of potential application for the further design of new antiretroviral compounds beyond the academic interest of the work?

4-. The quality of some images is not good enough and shows faded and blurred areas (ie., some Western blots such as those depicted in Fig.1). Please, increase quelity of images.

Have the images been edited in any way before mounting in the corresponding Figures? Please, indicate if some editing has been performed.

5-. Page 4, line 163. A supplemental Table (Table S1) is referred, but no such table appears in the supplementary material provided. Please revise it.

6-. Some additional information regarding the identity of the targeted fusion proteins would be advisable. How do you isolate or purify? Some details regarding the structural identification and purity of the molecules could be of interest if available.

Final considerations:

The lifecycle of the HIV-1 in host cells is a complex, multifactorial and subtly regulated process. At the present moment, the biological functions and the role of the proteins of the pathogen have been identified and largely elucidated. However, in spite of the extensive progress made in recent years, there are several aspects still to be unravelled. Particularly, clarifying the mechanisms of the pathogen involving interaction with hosts factors required for many viral processes is an actual challenge for researchers. These findings are needed for the development of novel antiviral agents and therapies.

The authors carried out an impressive effort in terms of number and complexity of different complementary techniques directed to determine the factors governing the effects of a series of fusion molecules containing ESCRT-III proteins and ns3 protease (from HCV). This is a preliminary (but exhaustive) study on a non-sufficiently explored stage of the HIV-1 lifecycle. All the experiments have been properly designed and performed. The description and discussion of the experiments carried out and the explanation of the results obtained are coherent and, when required to further clarify or corroborate some aspects, complementary experiments were proposed and performed.

Graphical results are correctly presented. It must be highlighted the proper statistic treatment, which helps to determine and quantify the significance of the measurements. Microscopy images are clear and self-explanatory.

The manuscript is properly written and described/discussed, in a very comprehensive manner. I think the proposed manuscript could be of interest for the specialised reader and deserves an opportunity to be considered for publication if authors take into consideration the few suggestions proposed by the reviewer (listed above). If authors provide reasonable responses to the points issued, the editorial office could give the opportunity to the authors of a resubmission.

Comments on the Quality of English Language

I am not qualified to assess the quality of English in this paper as I am nor english native speaker. However, I did nor detect any idiomatic conflicts (it seems to me very good written). 

Author Response

 We would like to thank the reviewers for its very thoughtful and thorough review of our manuscript. Please find the detailed responses below and the corresponding revisions/corrections highlighted in red track changes in the re-submitted files.

1-. In the introduction section, page 2, the paragraph contained between lines 78 (“We find that…” ) and 83 (…Co-localizatio”) actually anticipates results of this work. Consequently, I would suggest to move forward to the end of the paper (i.e., page 16, lines 479 onwards).

2-. In parallel, in page 4, paragraph located between lines 158 (”to study the…) and 167 (…proteins (Figure 1a)” should be located before as anticipates the aim and objective of the research. Consequently, I would suggest to move forward to the beginning of the paper (i.e., page 2, lines78 onwards). Figure 1a would be split from the original Figure and relocated accordingly on page 2.

This change would help to understand the aim and objective of the research prior to the description of the results as currently deployed.

We concur with the reviewer's suggestion that relocating the description of the tools used to the introduction would enhance clarity. Accordingly, we have made the necessary modifications to our manuscript.

3-. Authors Would indicate the interest and projection of this study focused on ESCRT-III inhibition.

Is ESCRRT-III a validated target against HIV-1?

What is the projection of the work thinking in a close-time therapeutic application against HIV-1 infection?

Are the described fusion proteins clinically usable or just a research tool? 

Is ESCRT-III drugable?

What is the interest/projection of this study in terms of potential application for the further design of new antiretroviral compounds beyond the academic interest of the work?

 We appreciate the reviewer for placing our tool in a broader context. However, it's essential to note that our assay exerts an inhibitory action without completely abrogating HIV-1 budding. Additionally, the ESCRT-III machinery catalyzes various membrane remodeling processes beyond HIV-1 budding, including multi-vesicular endosome formation, exosome biogenesis, peroxisome and recycling endosome fission, cytokinesis, nuclear envelope reformation, membrane repair, and autophagy. Therefore, ensuring the safety of drugs or dominant negative tools targeting the ESCRT-III pathway is crucial before considering therapeutic applications.

Our assay is specifically designed for studying the in-cellulo ESCRT-III machinery during HIV-1 budding. It provides a direct means to transiently block ESCRT-III filaments, enabling the pursuit of high-resolution imaging methods. This approach would offer invaluable structural insights into native ESCRT-III complexes and their pivotal role in the membrane fission process.

4-. The quality of some images is not good enough and shows faded and blurred areas (ie., some Western blots such as those depicted in Fig.1). Please, increase quelity of images.

To address the reviewer's comment, we have added all blot images in Supplementary Figure S1.

Have the images been edited in any way before mounting in the corresponding Figures? Please, indicate if some editing has been performed.

The images have not undergone editing; only brightness and contrast adjustments have been made to enhance clarity.

5-. Page 4, line 163. A supplemental Table (Table S1) is referred, but no such table appears in the supplementary material provided. Please revise it.

 Done

6-. Some additional information regarding the identity of the targeted fusion proteins would be advisable. How do you isolate or purify? Some details regarding the structural identification and purity of the molecules could be of interest if available.

We agree with the reviewer regarding the potential significance of structural information on CHMP-NS3-FP fusion. Notably, these proteins were not purified in vitro ; whole cell extract  were performed. Despite our efforts to utilize Alphafold for gaining structural insights, the validation of these models would require an extensive confirmation process, surpassing the scope of this article.

Final considerations:

The lifecycle of the HIV-1 in host cells is a complex, multifactorial and subtly regulated process. At the present moment, the biological functions and the role of the proteins of the pathogen have been identified and largely elucidated. However, in spite of the extensive progress made in recent years, there are several aspects still to be unravelled. Particularly, clarifying the mechanisms of the pathogen involving interaction with hosts factors required for many viral processes is an actual challenge for researchers. These findings are needed for the development of novel antiviral agents and therapies.

The authors carried out an impressive effort in terms of number and complexity of different complementary techniques directed to determine the factors governing the effects of a series of fusion molecules containing ESCRT-III proteins and ns3 protease (from HCV). This is a preliminary (but exhaustive) study on a non-sufficiently explored stage of the HIV-1 lifecycle. All the experiments have been properly designed and performed. The description and discussion of the experiments carried out and the explanation of the results obtained are coherent and, when required to further clarify or corroborate some aspects, complementary experiments were proposed and performed.

Graphical results are correctly presented. It must be highlighted the proper statistic treatment, which helps to determine and quantify the significance of the measurements. Microscopy images are clear and self-explanatory.

The manuscript is properly written and described/discussed, in a very comprehensive manner. I think the proposed manuscript could be of interest for the specialised reader and deserves an opportunity to be considered for publication if authors take into consideration the few suggestions proposed by the reviewer (listed above). If authors provide reasonable responses to the points issued, the editorial office could give the opportunity to the authors of a resubmission.

Reviewer 3 Report

Comments and Suggestions for Authors

The manuscript by Wang et al describes the development of an inducible ESCRT-III inhibition tool to control HIV-1 budding. The report describes a drug resistance transient inhibitory ESCORT-III system to help solve the challenge of transient nature of ESCORT-III function. The experiments are thorough and are straight forward with relevant controls. Such inducible ESCRT-III system will help obtain more insight into its structure and function ESCRT-III at HIV-1 budding sites to ESCRT dependent membrane remodeling processes.  All experiments, however, were performed in cells that are not HIV targets.  The manuscript would be enhanced if one simple experiment is included in T cells or other HIV target cells.  A simple discussion of whether endogenous (if there are) ESCRT-III protein has any influence on their system would also help

Author Response

We would like to thank the reviewer for taking the time to review this manuscript. Please find the detailed responses below and the corresponding revisions/corrections highlighted in red track changes in the re-submitted files.

We agree with the reviewer that our assay was performed in cell lines that are not HIV-1 targets. The rationale for this choice was the use of adherent cells, enabling comprehensive cellular imaging. T cells, being round, would not be suitable for our imaging purposes.

Endogenous ESCRT-III affects the system and it is expected that endogenous ESCRT-III proteins will co-polymerize with the CHMP-fusion proteins described here. As discussed, the fusion proteins will likely affect the remodeling of the polymers by VPS4 as it is expected for classical dominant negative forms of ESCRT-III proteins that can inhibit virus release completely.

Round 2

Reviewer 3 Report

Comments and Suggestions for Authors

 The response to the comment performing experiments in HIV target cells was 'the rationale for this choice was the use of adherent cells, enabling comprehensive cellular imaging. T cells, being round, would not be suitable for our imaging purposes.". This response is partial and could have performed a simple supportive experiment.  However, as pointed out by this reviewer, previously,  the cell lines tested could serve as a proof principle.